# Potential of Thai Herbal Extracts on Lung Cancer Treatment by Inducing Apoptosis and Synergizing Chemotherapy

**DOI:** 10.3390/molecules25010231

**Published:** 2020-01-06

**Authors:** Juthathip Poofery, Patompong Khaw-on, Subhawat Subhawa, Bungorn Sripanidkulchai, Apichat Tantraworasin, Somcharoen Saeteng, Sopon Siwachat, Nirush Lertprasertsuke, Ratana Banjerdpongchai

**Affiliations:** 1Department of Biochemistry, Faculty of Medicine, Chiang Mai University, Chiang Mai 50200, Thailand; juthathip_p@cmu.ac.th (J.P.); patompong_k@cmu.ac.th (P.K.-o.); subhawat_s@cmu.ac.th (S.S.); 2Center for Research and Development of Herbal Health Products, Faculty of Pharmaceutical Sciences, Khon Kaen University, Khon Kaen 40002, Thailand; bungorn@kku.ac.th; 3Clinical Epidemiology and Clinical Statistic Center, Faculty of Medicine, Chiang Mai University, Chiang Mai 50200, Thailand; apichat.t@cmu.ac.th; 4Department of Surgery, Faculty of Medicine, Chiang Mai University, Chiang Mai 50200, Thailand; tengearneae@gmail.com (S.S.); sophon.s@cmu.ac.th (S.S.); 5Department of Pathology, Faculty of Medicine, Chiang Mai University, Chiang Mai 50200, Thailand; nlertpra@hotmail.com

**Keywords:** *Bridelia ovata*, *Croton oblongifolius*, *Erythrophleum succirubrum*, herbal extract, lung cancer, anti-tumor, synergism, apoptosis, primary cells

## Abstract

The incidence of lung cancer has increased while the mortality rate has continued to remain high. Effective treatment of this disease is the key to survival. Therefore, this study is a necessity in continuing research into new effective treatments. In this study we determined the effects of three different Thai herbs on lung cancer. *Bridelia ovata*, *Croton oblongifolius*, and *Erythrophleum succirubrum* were extracted by ethyl acetate and 50% ethanol. The cytotoxicity was tested with A549 lung cancer cell line. We found four effective extracts that exhibited toxic effects on A549 cells. These extracts included ethyl acetate extracts of *B. ovata* (BEA), *C. oblongifolius* (CEA), and *E. succirubrum* (EEA), and an ethanolic extract of *E. succirubrum* (EE). Moreover, these effective extracts were tested in combination with chemotherapeutic drugs. An effective synergism of these treatments was found specifically through a combination of BEA with methotrexate, EE with methotrexate, and EE with etoposide. Apoptotic cell death was induced in A549 cells by these effective extracts via the mitochondria-mediated pathway. Additionally, we established primary lung cancer and normal epithelial cells from lung tissue of lung cancer patients. The cytotoxicity results showed that EE had significant potential to be used for lung cancer treatment. In conclusion, the four effective extracts possessed anticancer effects on lung cancer. The most effective extract was found to be *E. succirubrum* (EE).

## 1. Introduction

Lung cancer has been the leading cause of cancer-related deaths for many years and incidence and mortality statistics vary widely worldwide [1]. In both sexes combined, lung cancer is the most diagnosed cancer (11.6% of the total cases) and the leading cause of cancer death (18.4% of the total cancer deaths) worldwide [2]. Tobacco consumption is a major risk factor for lung cancer. Other factors include genetic susceptibility, diet, alcohol consumption, occupational exposures, and air pollution [3]. Indoor and outdoor air pollution create a high-risk for lung cancer in people living in several regions of Asia, including Thailand [4,5]. Surgical resection is the primary treatment of early stage lung cancer. For advanced disease (stage IIIB and stage IV) chemotherapy, radiotherapy, targeted therapy, and immunotherapy are the principle treatments, however, the 5-year survival rate of these standard treatments is only around 15% and metastasis is still one of the major causes of death. There are two main subtypes of lung cancer: Non-small cell lung cancer (NSCLC) and small cell lung cancer (SCLC). Around 80% of all lung cancers are determined to be NSCLC. This type of lung cancer is usually diagnosed at an advanced stage of metastasis. At this stage the possibility of surgery having a curative effect is lessened greatly. [6]. Moreover, drug resistance is a major cause for therapeutic failure in NSCLC leading to tumor recurrence and disease progression [7]. 

Several anticancer agents have been found in natural products which have been investigated and developed to become effective chemotherapeutic cancer drugs [8]. However, many of these chemotherapeutic drugs have been shown to produce significant toxic side effects and drug resistance. Thus, continued research needs to be pursued to find more effective natural products that provide fewer negative side effects [9]. In Thailand there are various herbs that possess medicinal potential for anti-cancer treatments. In this study ethyl acetate extracts and 50% ethanolic extracts from three plant species were examined for cytotoxic activity and apoptotic induction in a human lung cancer A549 cell line and were investigated the toxicity results in primary cancer cells obtained from human lung cancer tissues. The three herbal species studied were *Bridelia ovata* Decne, *Croton oblongifolius* Roxb., and *Erythrophleum succirubrum* Gagnep.

*B. ovata* and *C. oblongifolius* are in the Euphorbiaceae family. *B. ovata* is known as Ma-Ga in Thai and traditional medicine uses it as an expectorant, a laxative, and a medicinal astringent [10]. There are various phytochemicals in *B. ovata* that were previously identified as triterpenes and phytosterols [11,12]. A crude ethanolic extract of *B. ovata* was recently reported to inhibit human hepatocellular carcinoma HepG2 cell invasion and migration [13].

*C. oblongifolius*, or Plao-Yai in Thai, is a medicinal plant which acts as a tonic and a purgative and treats dysmenorrhea, dysentery, dyspepsia, and chronic liver enlargement. In addition, the combination of *C. oblongifolius* and *C. sublyratus* had been used as a treatment for gastric ulcers and gastric cancer in Thai traditional medicine [14]. The phytochemicals of *C. oblongifolius* have been reported to include megastigmane glycosides [15], diterpenoids such as labdanes [16,17], clerodanes [18,19,20], halimane [21], and cembranes [22,23,24]. Croblongifolin, the clerodane-type compound, exhibits cytotoxicity to human cancer cell lines, including HepG2, SW620, CHAGO, KATO3, and BT474 [19]. 

*E. succirubrum* belongs to the Leguminosae-Caesalpinioideae family. It is known as ‘Phan-Saat’ in Thai and is used to treat fever and skin diseases in Thai traditional medicine [25]. The cassaine diterpenoid dimers, which are isolated from the bark of *E. succirubrum*, have been reported to possess anti-cancer activity by the induction of apoptosis in human gastric adenocarcinoma cells [26]. Furthermore, the crude ethanolic extract of *E. succirubrum* exhibits moderate cytotoxicity against human hepatocellular carcinoma cells (HepG2). However, the mechanism(s) of cell death is still elusive [25].

Apoptosis, the well-known cell death mechanism, is induced by many chemotherapeutic agents. Membrane blebbing, nuclear condensation, and apoptotic bodies are unique morphology characteristics of apoptotic cells that occur without cell inflammation [27]. There are two main pathways in apoptotic signaling. The first is the intrinsic pathway which is induced by intracellular stimuli such as DNA damage or oxidative stress. The Bcl-2 family is a protein family composed of pro-apoptotic and anti-apoptotic proteins which tightly regulate the intrinsic pathway via the mitochondria. During apoptosis induction, the pro-apoptotic proteins (Noxa, Puma, Bax, and Bak) are upregulated to inhibit the function of anti-apoptotic proteins (Bcl-2, Bcl-xl, and Mcl-1), and induce the mitochondrial outer membrane premiumization (MOMP). This causes intermembranous space protein release and mitochondrial transmembrane potential loss [28]. Then, caspase 9 and caspase 3 are activated to induce cell apoptosis. The other pathway is the extrinsic pathway which is induced by death ligand-receptor binding on the cell membrane. The oligomerization of the receptors induces the formation of a protein complex in the cytosol which activates caspase 8 and caspase 3 and then induces apoptosis [29]. 

## 2. Results

### 2.1. Cytotoxicity Test of the Extracts Against Lung Cells

Three ethyl acetate extracts (BEA, CEA, and EEA) and three 50% ethanolic extracts (BE, CE, and EE) were examined for cytotoxicity against an A549 human lung cancer cell line by MTT assay. At 24 h incubation, the percentages of cell viability of A459 cells were significantly decreased at a dose dependent manner by treatment with BEA, CEA, EEA, and EE extracts, but not with BE and CE. Also, BEA, CEA, EEA, and EE decreased the A549 cells viability in a time dependent manner (24, 48, and 72 h), but not with BE and CE (Figure 1). Therefore, these four effective extracts (BEA, CEA, EEA, and EE) were used to determine their cytotoxicity against peripheral blood mononuclear cells (PBMCs). The results in Figure 2 showed a significant toxic effect when the cells were treated with CEA but not with BEA, EEA, and EE after treating the cells for 24 h. However, at 48- and 72-h treatments the effective extracts significantly decreased the PBMCs viability. So, 24 h treatment was used for further experiments as the least toxic on the PBMCs but still toxic on cancer cells. The IC values against A549 and PBMCs, and selectivity index (SI) of the effective extracts are shown in Table 1. The most effective inhibitory herbal extracts were EE > CEA > EEA > BEA, concerning the IC_50_ values at 24 h treatment.

### 2.2. Identification of the Phytoconstituents in the Effective Extracts by Gas Chromatography–Mass Spectrometry (GC-MS)

The chromatogram in Figure 3 showed the GC-MS results of BEA, CEA, EEA, and EE, including the structures of the three main compounds in each extract. The BEA extract contained 28.99% of friedelan-3-one, 22.52% of stigmast-5-en-3-ol, and 9.09% of palmitic acid. The main compound in the CEA extract was kaur-16-en-18-oic acid (50.24%), whereas the other two main compounds were stigmast-5-en-3-ol and benzoic acid, which were only at 3.60% and 3.24%, respectively. Even though EEA and EE were extracted from the same plant their main compounds are different. EEA extract contained 22.74% of oleic acid, 12.89% of palmitic acid, and 11.73% of pyrogallol, whereas EE extract contained 37.82% of pyrogallol, 21.75% of ethyl gallate, and 6.26% of resorcinol. Data from the active compounds in the effective extracts enabled the identification of 26 compounds from BEA, 13 compounds from CEA, 22 compounds from EEA, and 17 compounds from EE. Retention time, compound name, and the percentages of each compound in the BEA, CEA, EEA, and EE are shown in Table 2, Table 3, Table 4 and Table 5, respectively. 

### 2.3. Combined Effects of the Effective Extracts with the Chemotherapeutic Drugs

The conventional chemotherapeutic drugs, etoposide (ETS) and methotrexate (MTX), were used as treatment on the A549 cells alone or in combination with the effective extracts. The cells viability was determined by MTT assay. The cytotoxicity results, concentrations of the effective synergy extracts and chemotherapeutic drugs, and CI values were shown in Figure 4A,C,E). The cell viability results from the chemotherapy drugs alone showed that the A549 cells tended to resist the MTX but responded to ETS at a high dose. However, the combination of MTX with BEA, MTX with EE, and ETS with EE could reduce the cell viability to be lower than drug treatment alone and displayed the synergistic effect at some concentration. The percentage of cell viability was used for calculating the fractional inhibition (Fa) and combination index (CI) by the Chou-Talalay method. The Fa-CI plots or Chou-Talalay plots for drug combination at the Figure 4B,D,F allow quantitative determination of drug interactions, where CI <1, =1, and >1 indicates synergism, additive effect, and antagonism, respectively [30]. The interpretation of CI value was shown in Table 6 which related to the results in Figure 4.

### 2.4. Apoptotic Induction by the Effective Extracts

The morphology of the A549 cells after treatment with the effective extracts was detected by PI staining using fluorescence microscopy. The number of cells with condensed nuclei and apoptotic bodies increased in a dose dependent manner for all of the effective extracts (Figure 5). In addition, the results from flow cytometer of the annexin V-FITC/PI assay showed a significant increase in the percentage of apoptosis cells for every effective extract as shown in Figure 6. This indicates that the effective extracts induced A549 cell apoptosis.

### 2.5. Reduction of Mitochondrial Transmembrane Potential (∆Ψm)

Mitochondria are the main organelles which are affected when the cell undergoes apoptosis [31]. Dysfunction of mitochondria, which occurs during apoptosis, was determined by the reduction of ∆Ψm using DiOC_6_ fluorescence dye staining which produces a decreasing intensity with ∆Ψm loss. The results in Figure 7 demonstrated that the effective extracts induced ∆Ψm loss in A549 cells in a dose-dependent manner. This indicates that the A549 cell apoptosis was mediated via mitochondrial dysfunction.

### 2.6. Intracellular Reactive Oxygen Species (ROS) Generation

Production of ROS at high levels induces mitochondrial dysfunction, cell damage, and apoptosis [32]. The results of the intracellular ROS, which were measured by DCFH-DA in Figure 8, showed that the ROS generation significantly increased after the cells were treated with the effective extracts. This indicates that the mitochondrial signaling pathway was activated by the effective extract treatments on the A549 cells.

### 2.7. Induction of Noxa in Both Gene and Protein Levels

Noxa is a member of the pro-apoptotic Bcl-2 family (BH3-only protein) which is upregulated during apoptosis [33]. Gene and protein levels of Noxa were determined by real-time RT-PCR and Western blot analysis, respectively. After the A549 cells were treated with the effective extracts, the results showed that both gene and protein levels were significantly increased for every effective extract (Figure 9 and Figure 10). Therefore, the effective extracts upregulated Noxa to induce mitochondria-mediated apoptosis in A549 cells.

### 2.8. The Effect of the Effective Extracts on Primary Lung Cells

We examined the cytotoxicity of each of the effective extracts and the conventional chemotherapy drugs (doxorubicin; DOX, etoposide; ETS, and vinblastine; VBT) against primary lung cancer cells and normal lung cells which were obtained from lung cancer patients. The patients’ characteristics are shown in Table 7. Nine samples were successfully cultured as primary cancer cells, whereas seven samples were cultured as primary normal cells. The cytotoxicity results from the MTT assay was used for calculating the IC_50_ of each substance (extracts and chemotherapeutic drugs) on sensitive samples as shown in Table 8. The sensitive sample and resistant sample were separated by the percentages of cell variability. The cell samples where the percentage of cell viability were reduced to less than 50% are considered to be the sensitive samples for each substance treatment. On the other hand, cell samples which possessed cell viability over 50% were identified as the resistant samples. The results showed that all of the primary lung cancer samples were sensitive to CEA and EE, but only three cancer samples were resistant to BEA and only two cancer samples were resistant to EEA. For the chemotherapy drugs, there were six sensitive samples for every drug, but they were not the same samples of each drug. In addition, most normal samples were sensitive to every substance except BEA, which was toxic to only two normal samples. The *p*-values in Table 8 show that the IC_50_ of EE against lung cancer samples was significantly lower than normal lung samples (*p*-value < 0.05). As shown in Table 9, the mean difference of each substance demonstrates both quantity and direction and was compared between lung cancer samples and normal lung samples. Only the EE treatment showed an IC_50_ (at 4.38 µg/mL) against lung cancer samples that is significantly less than the normal lung samples with 95% confidence.

## 3. Discussion

Herbs are the main sources for drugs. Many herbs are used in traditional medicine and have been developed to become therapeutic drugs [34]. *B. ovata*, *C. oblongifolius*, and *E. succirubrum* are Thai herbs in the plant genetic conservation project under the Royal Initiative of Her Royal Highness Princess Maha Chakri Sirindhorn (RSPG). This project aims to develop plant genetic resources for the maintenance of plant varieties, and for the potential to develop future advances for the farming and business sector in Thailand. In this study we investigated the effects of these Thai herbal extracts in order to analyze their potential use in future alternative treatments for lung cancer.

Since a major subgroup of lung cancer is NSCLC, A549 cells were used in this study. The A549 cell line is made up of adenocarcinoma from human alveolar basal epithelial cells which are commonly used as a model for the study of lung cancer and for the development of drug therapies [35]. This study began with a cytotoxicity test of three plants which were extracted with two different solvents, ethyl acetate and 50% ethanol. Six extracts were tested against A549 cells. The cytotoxicity results demonstrated that BEA, CEA, EEA, and EE extracts exhibited toxicity towards A549 cells at 24 h incubation. Although these four effective extracts were toxic to PBMCs, the IC_50_ of PBMCs was significantly greater than that of A549 cells. Therefore, the concentrations of each effective extract used for treating cancer cells are not toxic to normal PBMCs. The SI values of these four effective extracts were more than three, which indicates a high selectivity of the effective extracts tested on the A549 cells [36,37].

The GC-MS chromatograms of the effective extracts showed many different peaks for the active compounds. Only the structures of the three most abundant compounds were displayed in Figure 4. Even though *B. ovata* and *C. oblongifolius* are plants from the same family (Euphorbiaceae) and phytosterols were identified in both BEA and CEA (campesterol, stigmasterol, and stigmast-5-en-3-ol), the main compounds in these plants are different. Friedelan-3-one (28.99%) and Stigmast-5-en-3-ol (22.52%) are the main compounds in BEA. Friedelan-3-one, or friedelin, is a triterpenoid which has been reported to have anti-tumor activities against various cancer types, including human malignant melanoma A375, human cervical tumor HeLa, human macrophage tumor THP-1, and mouse lung epithelial tumor L929 cells [38]. The inhibitory effect of friedelan-3-one on VEGF-induced Kaposi’s sarcoma cell proliferation via apoptotic cell death induction [39] and on breast cancer MCF-7 cell growth associated with p53 and caspase activation [40] were also reported. Stigmast-5-en-3-ol is the second most abundant compound in both BEA (22.52%) and CEA (3.60%). The apoptotic and antiproliferative effects against human leukemia HL-60 and MCF-7 cells were also demonstrated [41].

However, the main compound of CEA is kaur-16-en-18-oic acid, which makes up 50.24% of the total constituents in CEA but was not found in BEA. This may be the reason that CEA possessed more effective activity comparing with BEA. Kaur-16-en-18-oic acid is a kaurane-type diterpenoid which has anticancer potential [42]. It has been reported that kaur-16-en-18-oic acid induces apoptosis cell death in HL-60 [43], HeLa and CaSki cervical cancer cell lines [44], including a significant anti-cancer effect against an A549 cell line [45].

The most effective extract against A549 cells and the only extract that had a significant effect on primary lung cancer cells was EE. Although the extracts of *E. succirubrum* (both ethyl acetate and ethanol) are effective on A549 cells, EE remarkably produced a more significant effect than EEA. The main compounds in EE are pyrogallol (37.82%) and ethyl gallate (21.75%). Pyrogallol, a catechin compound, is known as superoxide anion generator. It induces GSH depletion-mediated cell death in several cell types including lung cancer cells [46]. Previous studies have reported that pyrogallol has highly cytotoxic effects on human lung cancer cell lines. Cell growth of A549, squamous cell lung carcinoma H520, and lung adenocarcinoma H441 cells were inhibited by pyrogallol via cell cycle arrest [47,48]. In addition, pyrogallol inhibits human lung adenocarcinoma Calu-6 cell via caspase-dependent apoptosis and cell cycle arrest [49,50]. Moreover, ethyl gallate, which is a phenolic compound, has been reported to exhibit anticancer activity against many cell lines such as human prostate cancer PC3, cervical cancer HeLa and CaSki, human hepatocellular carcinoma Hep3B and HepG2 cells [51]. In previous studies ethyl gallate induces HL-60 apoptosis via the mitochondria-mediated pathway. It also suppresses proliferation and invasion of breast cancer via Akt-NF-kB signaling, and inhibits patient-derived esophageal tumor growth in an in vivo mouse model via ERK1/2 inhibition [52,53,54]. Therefore, the two most abundant compounds, pyrogallol and ethyl gallate, might possess significant anticancer activity in EE.

EEA contains oleic acid (22.74%) and palmitic acid (12.89%) which are the two most abundant compounds. They are well known fatty acids which have unclear activities on cancer cells. There are some reports about oleic acid that demonstrate that they possess anti-tumor activity depending on the type of cancer, and are linked to the inhibition of angiogenesis [55], metastasis [56], and apoptosis induction [57]. Alternatively, oleic acid has been reported to promote cell proliferation and migration in aggressive metastatic cancer cells via enhanced β-oxidation mediated by AMPK activation [58]. There are similar issues in palmitic acid which has been reported to reduce and induce cancer cell growth [59,60]. The third most abundant compound in EEA is pyrogallol (11.73%) which is also the most abundant compound in EE.

Currently, chemotherapy is the main treatment for lung cancer and is a palliative remedy for patients. However, some patients do not respond well. This generally leads to an increase in the dosage of drugs which increases the potential for adverse side effects. Moreover, some patients are resistant to chemotherapy. Therefore, a combination of chemotherapy is commonly used for cancer treatment [61]. The main purpose of using drug combination therapies is to create a synergistic therapeutic effect that allows for a reduction of the dosage of the chemotherapy drugs. This potentially lessens adverse side effects while reversing drug resistance [62]. The level of synergism is quantified by the drug combination index (CI) which is calculated by the Chou and Talalay’s method [63]. The CI offers a quantitative definition for the additive effect (CI = 1), synergism (CI < 1), and antagonism (CI > 1) in drug combinations. In this study, A549 cells were treated with carboplatin, methotrexate, vinorelbine, or etoposide alone, then treated with a combination of each drug with each effective extract, but the results show only the synergistic combination. We found three synergistic combinations, which are BEA with MTX (Figure 3A,B), EE with MTX (Figure 3C,D), and EE with EPS (Figure 3E,F). A549 cells did not respond well to EPS (Figure 3E) and were resistant to MTX (Figure 3A,C). The combinations of MTX with BEA and MTX with EE showed synergistic effects at low concentrations of MTX. Therefore, EE and BEA can sensitize A549 cells to MTX. However, a high concentration of MTX (250 µg/mL) can increase the CI value to more than one in both combinations with BEA (CI = 1.12653) and EE (CI = 1.42557). This means that the combination treatments possess a level of antagonism if used with a high dose of MTX. Therefore, it is more effective to treat A549 cells with lower concentrations of MTX combined with BEA or EE. These combination treatments can reduce the dose of MTX and feasibly decrease side effects in further clinical studies.

The mode of cancer cell death was determined by fluorescence microscopy and flow cytometry. Although there are a number of molecular markers for the cell death mechanism, morphological criteria is still the standard for defining the mode of cell death [64]. PI staining was used for nuclear change observation. The results show apoptotic morphology with bright red, condensed nuclei (intact or fragmented). The formation of apoptotic bodies was compared to the control cells that are displayed as round, intact, red nuclei (Figure 5A). Apoptotic cells and apoptotic bodies increased in a dose-dependent manner compared to the control after treating the cells with these four effective extracts. The quantification of apoptosis was obtained by flow cytometry. During apoptosis, phosphatidylserine (PS) becomes exposed on the outside of the membrane. The detection of PS by annexin V-FITC is used for the estimation of the incidence of both early (Annexin V^+^/PI^−^) and late (Annexin V^+^/PI^+^) apoptosis [65]. The flow cytometry results confirmed that the effective extracts induced A549 cell apoptosis (Figure 6). Therefore, apoptosis cell death was the mode of cell death that was triggered by these four effective extracts.

Mitochondrial disturbances occurred during apoptosis in several ways, including through a loss of ∆Ψm, a release of apoptotic proteins from the intermembranous space into the cytosol, and a generation of ROS [66]. A reduction of ∆Ψm was determined and it was found that the percentage of cells with a loss of ∆Ψm increased after treatment of A549 with the effective extracts. Moreover, ROS generation increased in extract-treated cells compared to the control cells. This suggests that the effective extracts induced A549 apoptosis, which involved a disruption of the mitochondria and intracellular oxidative stress.

Apoptotic pathways require several protein functions. The Bcl-2 family proteins play important roles in the apoptosis pathway. The two main types of these proteins are pro-apoptotic and anti-apoptotic. There are two different subgroups of pro-apoptotic protein, activator and effector. The BH3-only activators promote the multidomain effector forming of pores on the mitochondrial outer membrane leading to ∆Ψm loss and initiating the apoptotic programmed cell death [67]. Noxa, a BH3-only activator, was examined for the upregulation of both gene and protein expression levels to indicate the ∆Ψm loss and cell apoptosis resulting from the function of the Bcl-2 family. The effective extracts induced ROS generation and Noxa upregulation in A549 cells leading to a loss of ∆Ψm and apoptotic cell death via the mitochondrial pathway.

Human cancer-derived cells are elementarily used as a model to study cancer biology and to analyze the efficacy of therapeutic anti-cancer agents. These cells are effective because they can be easily cultured and they have a long lifespan [68]. However, cancer cells possess a high heterogeneity. Cancer cell lines may have a limitation in representing these complicated cancer diseases [69]. Primary cancer cells are used to mimic the in vivo cancer cells better than cancer cell lines. These compounds have more predictive values and can be more useful in new drug discovery [70,71]. In this study we established both primary lung cancer cells and normal lung cells were obtained from the same patients in a cultured model in the mimicry of in vivo. The normal histology and cancer tissue pathology were proved by a pathologist from the Department of Pathology, Maharaj Nakorn Chiang Mai Hospital, Faculty of Medicine, Chiang Mai University, Chiang Mai, Thailand. The patients’ characteristics are shown in Table 7. The molecular test of EGFR mutation predicted the response rates of tyrosine kinase inhibitors (TKIs) treatment in the lung cancer patients. The overexpression of EGFR has been reported in many cancers including NSCLC, which leads to cell proliferation and anti-apoptosis [72]. Therefore, the studies of a subset of NSCLC patients with EGFR mutant tumors, and initial therapy with TKIs appears to be a significant survival advantage [73]. Greater than 90% of EGFR mutation in NSCLC occurs as short in frame shift deletions in exon 19 or as point mutations in exon 21 [74]. In this study there were three patients with EGFR wild types, three patients with EGFR mutations and another three with an unknown EGFR status as shown in Table 7.

The primary cells were collected and were determined as either sensitive or resistant samples towards the inhibitory growth effect of lung cancers and normal lung cells. This indicated the effectiveness of each substance at IC_50_. The number of sensitive samples showed that CEA and EE were effective against cancer samples from every patient, but no sample was sensitive to BEA and EEA. The IC_50_ values in cancer cells showed that EE was the most effective extract (lowest IC_50_) in primary lung cancer cells, which corresponded to the A549 cell line result in this study. The IC_50_ of the substances against primary cells have a wide range of SD because the primary cells possessed various considerable heterogeneous properties, such as the expression levels of oncogenes, tumor suppressor genes, and multi-drug resistance genes. Moreover, comparing IC_50_ values of cancer and normal samples using the *p*-value in Table 8 demonstrated that the IC_50_ of EE against cancer samples was significantly different from that of the normal samples (*p* < 0.05). However, the data in Table 8 is a univariate analysis without confounding variable control and univariate analyses can yield misleading results. In this case, the multivariate analysis was more appropriate [75]. The multi-level random effects model was used for calculating the mean difference of IC_50_ comparing cancer samples and normal samples. Mean differences of IC_50_ for EE, and Doxorubicin and Vinblastine against cancer were less than those of normal samples (negative values) but only the *p*-value of EE (*p*-value = 0.04) exhibited a significant result (*p* < 0.05). There are many factors in the primary culture which made the results hard to be interpreted and repeated. The primary cell culture is difficult to establish without contamination. It is hard to obtain and has a limitation for cultured cell passages. Hence, there was no further examination for the modes of cell death in the primary cancer cells or in the normal lung cells due to such limiting factors.

In this study, EE exhibited the highest effect according to the lowest IC_50_ against A549 cell line and primary lung cancer cells. Also, it induced the synergistic effect in combined treatment with ETS or MTX. However, the effect of EE in apoptosis induction and ∆Ψm reduction was less effective than the other three effective extracts, whereas EE induced ROS generated around 1.5-fold of control which was the same as the other three extracts. This suggests the cytotoxicity of EE may be the result of EE induced in other mechanism(s) over mitochondrial apoptosis. Although, *Noxa* gene expression was at different levels between each extract, the protein which is the functional molecule, was increased in the similar level in each extract. This indicates Noxa might not be the key different effect of these four effective extracts. Therefore, more details of the mechanisms inducing lung cancer cell death by each extract need to be investigated in further studies.

## 4. Experimental Sections

### 4.1. Plant Materials

Twigs of the three herbs were collected and authenticated by Professor Dr. Bungorn Sripanidkulchai, Center for Research and Development of Herbal Health Products, Department of Pharmaceutical Chemistry, Faculty of Pharmaceutical Sciences, Khon Kaen University, Khon Kaen, Thailand. The voucher numbers are TT-OC-SK-1253, TT-OC-SK- 1215, and TT-OC-SK-1082 for *B. ovata*, *C. oblongifolius*, and *E. succirubrum*, respectively. The herbs were washed, cut, dried, and mashed. The herb powders were macerated with 50% ethanol or ethyl acetate at room temperature for 72 h. Then, the plant macerations were filtered and centrifuged at 500× *g* for 10 min and the supernatant was collected. The supernatant extracts were concentrated under rotary evaporator.

### 4.2. Reagents

Dulbecco’s Modified Eagle Medium (DMEM), Roswell Park Memorial Institute (RPMI)-1640, fetal bovine serum (FBS), phosphate-buffered saline (PBS), trypsin-EDTA solution, penicillin, and streptomycin were purchased from GIBCO-Invitrogen (Carlsbad, CA, USA). Small Airway Epithelial Cell Growth Medium (SAGM) and its supplement; bovine pituitary extract (BPE), insulin, hydrocortisone, gentamicin-amphotericin (GA)-1000, retinoic acid, bovine serum albumin-fatty acid free (BSA-FAF), transferrin, triiodothyronine, epinephrine, and human epidermal growth factor (hEGF) were purchased from Lonza (Walkersville, MD, USA). Collagenase and elastase were obtained from Worthington Biochemical (Lakewood, NJ, USA). Histopaque-1077, 3-(4,5-dimethyl-2-thiazolyl)-2,5-diphenyltetrazolium bromide (MTT), 3,3′-dihexyloxacarbocyanine iodide (DiOC_6_), propidium iodide (PI), and 2′,7′-dichlorodihydrofluoresceine diacetate (DCFH-DA) were purchased from Sigma Chemica (St. Louis, MO, USA). Annexin V-FITC/PI kit was obtained from Roche, Mannheim, Germany. Tiangen RNA prep pure kit was purchased from Tiangen Biotech (Beijing, China). SensiFAST SYBR Lo-ROX kit from Bioline (Taunton, MA, USA). RevertAid first-strand cDNA synthesis kit and primers were obtained from Thermo Fisher Scientific (Waltham, MA, USA). Antibodies against Noxa and β-Actin, and HRP-conjugated secondary antibody were purchased from Abcam (Cambridge, UK). SuperSignal West Pico Chemiluminecent Substrate was obtained from Pierce (Rockford, IL, USA).

### 4.3. Cells Culture

A549 is the human adenocarcinoma cell line that was obtained from American Type Culture Collection (ATCC, Manassas, VA, USA). The cells were cultured in DMEM supplemented with 10% FBS, 100 Units/mL penicillin, and 100 µg/mL streptomycin. Cells were grown at 37 °C in a 5% CO_2_ atmosphere.

PBMCs were obtained from a buffy coat bag from volunteers at Blood Bank Unit, with the informed consent signed, whereas the approval was available from the Ethic Committee of the Maharaj Nakorn Chiang Mai Hospital, Faculty of Medicine, Chiang Mai University as the project code BIO-2558-03332. The cells in the buffy coat were separated by histopaque-1077 following density gradient centrifugation standard protocol. The PBMCs were cultured in RPMI-1640 medium and supplemented with 10% FBS, 2 mM glutamine, 100 Units/mL penicillin, and 100 µg/mL streptomycin at 37 °C in a 5% CO_2_ atmosphere [76].

### 4.4. Cytotoxicity Test

The cytotoxicity of the extracts and chemotherapy drugs was determined to analyze their inhibitory effects on A549 cells and PBMCs growth by MTT assay as previously described [77]. The cells were treated with the extracts and/or chemotherapy drugs at various concentrations for 24 h, and more 48- and 72-h treatment for the cytotoxicity screening of the extracts. The treatments used a final concentration of DMSO of less than 0.2% to avoid DMSO vehicle toxicity. The percentages of reduction in cell viability were compared to untreated control cells and then the inhibitory concentration (IC) values were calculated. The combined effects of the extracts and chemotherapeutic drugs were determined by the Chou-Talalay method on CompuSyn software using the cell viability results from MTT assay [62].

### 4.5. GC-MS Analysis

The GC-MS analysis of the effective extracts (BEA, CEA, EEA, and EE) were performed with Agilent technology GC 7890A coupled to Agilent technology MSD 5975C (EI) (Agilent, Santa Clara, CA, USA). The extracts were separated in a DB-5MS fused silica capillary column (30 m × 0.25 mm, 0.25 µm film thickness). The conditions of the GC were maintained operating at a temperature held at 50 °C for 3 min, and increased from 50 to 300 °C at a rate of 5 °C/min. The final isothermal was held for 20 min. The injector temperature was set at 250 °C with a 1.0 µL injection volume in 1:1 split mode. The detector temperature was set at 280 °C. Helium was used as a carrier gas with a 1 mL/min flow rate. The mass spectrometer was operated in electron impact mode at 70 eV and scanned from 50 to 550 amu. The compounds were identified by a comparison of the mass spectra and retention times through computer matching to the National Institute Standard and Technology (NIST) library (Gaithersburg, MD, USA) as well as with literature data.

### 4.6. Detection of Apoptotic Cell Morphology

Cell morphology was determined by PI staining, which stains the nuclei to observe the condensed nuclei and fragmented or apoptotic bodies. The cells were cultured on a coverslip and then treated with the effective extracts at IC_10_, IC_20,_ and IC_50_ for 24 h. The cells were stained with the PI method as previously described [78]. Then, the cells were examined under a fluorescence microscope (Olympus, Japan). A total of 200 cells (condensed nuclei and fragmented cells) per slide were recorded for apoptotic cells.

### 4.7. Apoptosis Determination Method

The Annexin V-FITC/PI assay is used to investigate cells which undergo apoptosis. Apoptosis cells are positive for annexin V (early apoptosis) or annexin V with PI (late apoptosis/necrosis) and are detected by a flow cytometer. After the cells were treated with the effective extracts at IC_10_, IC_20,_ and IC_50_ for 24 h, they were stained with annexin V-FITC and PI for 15 min. The stained cells were measured by a flow cytometer (Becton Dickinson, Frankin Lakes, NJ, USA) and analyzed with Cell Quest software program [79].

### 4.8. Assessment of Mitochondrial Depolarization

Depolarization of the ∆Ψm occurs during apoptosis. The changing of ∆Ψm is determined by DiOC_6_, which is a cationic fluorescence dye. The cells were treated with the effective extracts at IC_10_, IC_20,_ and IC_50_ for 24 h and then stained with DiOC_6_ at a final concentration of 40 nM for 15 min. Then, the cells were washed and analyzed by a flow cytometer (Becton Dickinson, Frankin Lakes, NJ, USA) [80].

### 4.9. Measurement of Intracellular ROS Generation

The intracellular ROS levels were measured by a fluorescence dye called DCFH-DA. After treatment with the effective extracts at IC_10_, IC_20,_ and IC_50_ and 0.3% of H_2_O_2_ (positive control) for 4 h, the cells were stained with DCFH-DA at a final concentration of 2 µM for 30 min. Then, fluorescence was detected using a fluorescence microplate reader, and was measured at 485 nm excitation and 525 nm emission wavelengths [81].

### 4.10. Gene Expression Analysis

After treatment of the cells with the effective extracts at IC_10_, IC_20_, and IC_50_ for 24 h, the total RNA was extracted using Tiangen RNAprep pure kit and then reverse transcribed to cDNA using a RevertAid first-strand cDNA synthesis kit. The mRNA expressions of *Noxa* were quantified by SensiFAST SYBR Lo-ROX on a 7500 Fast Real-time PCR system. The relative gene expression level was analyzed by 2^−∆∆Ct^, using *GAPDH* as a housekeeping gene. The primers of *Noxa* that were used: F-5′GCTGGAAGTCGAGTGTGCTA3′ and R-5′CCTGAGCAGAAGAGTTTGGA3′; and *GAPDH*: F-5′TGCACCACCAACTGCTTAGC3′ and R-5′GGCATGGACTGTGGTCATGAG3′ [82].

### 4.11. Protein Expression Analysis

The cells were treated with the effective extracts at IC_10_, IC_20,_ and IC_50_ for 24 h. Then, the proteins were determined at the concentration previously described [82]. Western blot analysis was conducted by the method described previously [78]. Briefly, protein extracts were subjected to 15% SDS-PAGE and blotted onto 0.45 µM nitrocellulose membranes which were incubated with Noxa antibody overnight. Then, after incubation with HRP-conjugated secondary antibody, protein bands were developed with chemiluminescence substrate and X-ray film exposure. The protein band intensity was analyzed by ImageJ software and β-actin was normalized and used as a protein loading control.

### 4.12. Lung Sample Processing

Normal lung tissue (*n* = 7) and cancer lung tissues (*n* = 9) were collected by Assistant Professor Apichat Tantraworasin, M.D., and his team of surgeons, General Thoracic Surgery Unit, Department of Surgery, Faculty of Medicine, Chiang Mai University, Chiang Mai, Thailand. The patients were informed and consent forms were signed before lung cancer tissues and normal tissues were obtained at the Operation Room, Department of Surgery, Maharaj Nakorn Chiang Mai Hospital, Faculty of Medicine, Chiang Mai University with the approval of the ethics committee referenced with the ethics number REC-25600901-11296 and project code BIO-2558-03332.

The lung tissues were immediately put into RPMI-1640 media and supplemented with 100 U/mL penicillin G and 100 µg/mL streptomycin. They were then transported to the lab. The blood clots, blood vessels and connective tissue were removed from the tissues, and then washed with sterile PBS twice. Next, they were cut into small pieces (~1 mm^3^) using a scalpel. The small pieces of tissues were incubated with collagenase at 50 units/mL and elastase at 10 units/mL at 37 °C and gently shaken every 15 min for 2 h. The digested tissues were passed through a 38 µm mesh sieve to collect the cells. Then, the cell suspension was washed twice by PBS and centrifuged at 300× *g* for 10 min. The cells were cultured in a Small Airway Epithelial Cell Growth Medium (SAGM) supplemented with bovine pituitary extract 2.0 mL, insulin 0.5 mL, hydrocortisone 0.5 mL, GA-1000 0.5 mL, retinoic acid 0.5 mL, bovine serum albumin-fatty acid free 5 mL, transferrin 0.5 mL, triiodothyronine 0.5 mL, epinephrine 0.5 mL, and hEGF 0.5 mL. The cultures were maintained at 37 °C in a humidified incubator with 5% CO_2_ [83,84]. The primary cells were cultured until passage 3 then the cells were treated with the effective extracted and chemotherapeutics drugs for 24 h and tested the cytotoxicity by MTT assay.

### 4.13. Statistical Analysis

The data was presented as a mean ± SD from three independent experiments. Statistical differences between the control (non-treatment) and treatment groups were determined by one-way ANOVA (Dunnett’s multiple comparisons test). The statistical significance is expressed as * *p* < 0.05, ** *p* < 0.01, *** *p* < 0.001. For primary cells the data in normal distribution was reported as a mean ± SD and was compared to the difference between cancer groups and normal groups by an unpaired t-test. Alternatively, non-normal distribution data was reported in median (interquartile range) and the statistical difference between groups was determined by a Wilcoxon rank-sum test. Lung cancer tissue and normal tissue were collected from the same patients. Therefore, the data was comparably correlated. The mean difference of the correlated data was analyzed by a multi-level random effects model which adjusted the IC_50_ by age, gender, cell type, and tumor size.

## 5. Conclusions

In summary, we investigated the potential of three Thai medicinal plants to be used for lung cancer treatment. *B. ovata*, *C. oblongifolius*, and *E. succirubrum* were extracted using ethyl acetate and ethanol as solvents to obtain BEA, CEA, and EEA and to get BE, CE, EE from all three herbs. Four of these extracts (BEA, CEA, EEA, and EE) induced human lung cancer cell apoptosis in the A549 cell line. Apoptotic induction occurred through the mitochondrial pathway which was evidenced by the ∆Ψm loss, ROS generation and upregulation of *Noxa* gene and Noxa pro-apoptotic protein. Moreover, the combination treatment between the extracts and conventional chemotherapeutic drugs showed a synergism of the three in combination, which were MTX with BEA, MTX with EE, and ETS with EE. We also investigated the cytotoxicity of the effective extracts against primary lung cancer cells. A significant result occurred when treating the cells with EE, the ethanolic extract of *E. succirubrum*. This was the most effective extract against both the A549 cancer cell line and the primary lung cancer cells. This suggests that further investigation should be further performed on this EE extract to explore its potential use as an anticancer agent in an in vivo model.

## Figures and Tables

**Figure 1 molecules-25-00231-f001:**
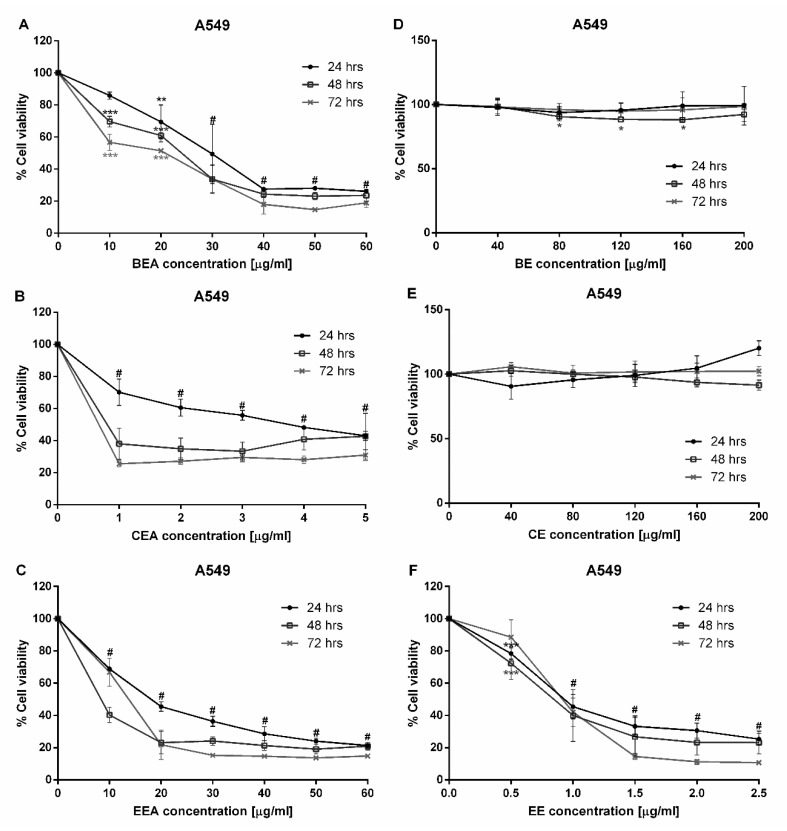
Cytotoxicity of the herbal extracts against the human lung cancer A549 cell line. Percent cell viability of A549 cells is shown on the *Y*-axis. The cells were determined after treatment with (**A**) BEA, (**B**) CEA, (**C**) EEA, (**D**) BE, (**E**) CE, and (**F**) EE for 24, 48, and 72 h at indicated concentrations as shown on the *X*-axis. The data are reported as mean ± SD of three independent experiments and was carried out in triplicate, * *p* < 0.05, ** *p* < 0.01, *** *p* < 0.001, and # *p* < 0.001 for all incubation times compared to the control condition.

**Figure 2 molecules-25-00231-f002:**
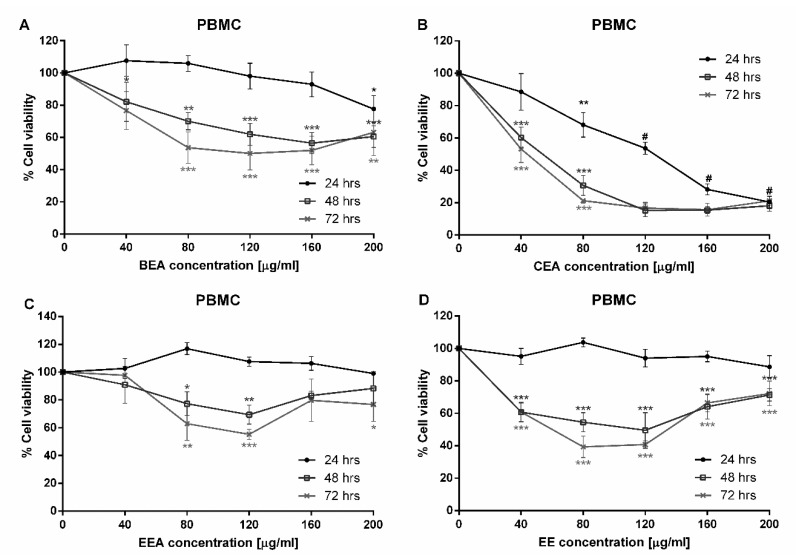
Cytotoxic effect of the effective extracts against PBMCs. PBMCs were separated and treated with the effective extracts; (**A**) BEA, (**B**) CEA, (**C**) EEA, and (**D**) EE for 24, 48, and 72 h at various concentrations on the *X*-axis. The data are reported as mean ± SD of three independent experiments carried out in triplicate, * *p* < 0.05, ** *p* < 0.01, *** *p* < 0.001 and # *p* < 0.001 for all incubation times compared to the control condition.

**Figure 3 molecules-25-00231-f003:**
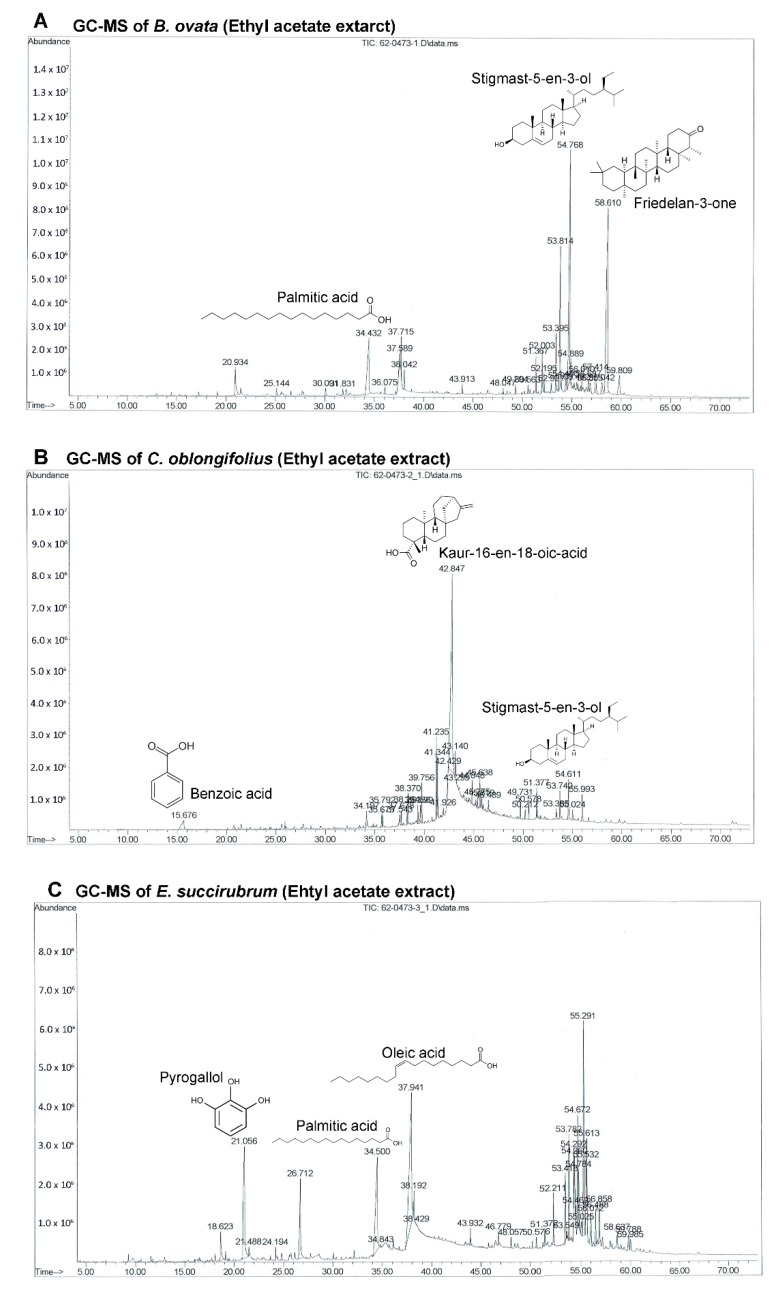
GC-MS chromatograms of the bioactive compounds are presented from the effective extracts. The four effective extracts were analyzed using Agilent Technology 7890A GC interfaced with Agilent technology 5975C (EI) MS. Their identification and characterization were based on their elution order in a DB-5MS column. The chemical structures of three major compounds in (**A**) BEA, (**B**) CEA, (**C**) EEA, and (**D**) EE are presented.

**Figure 4 molecules-25-00231-f004:**
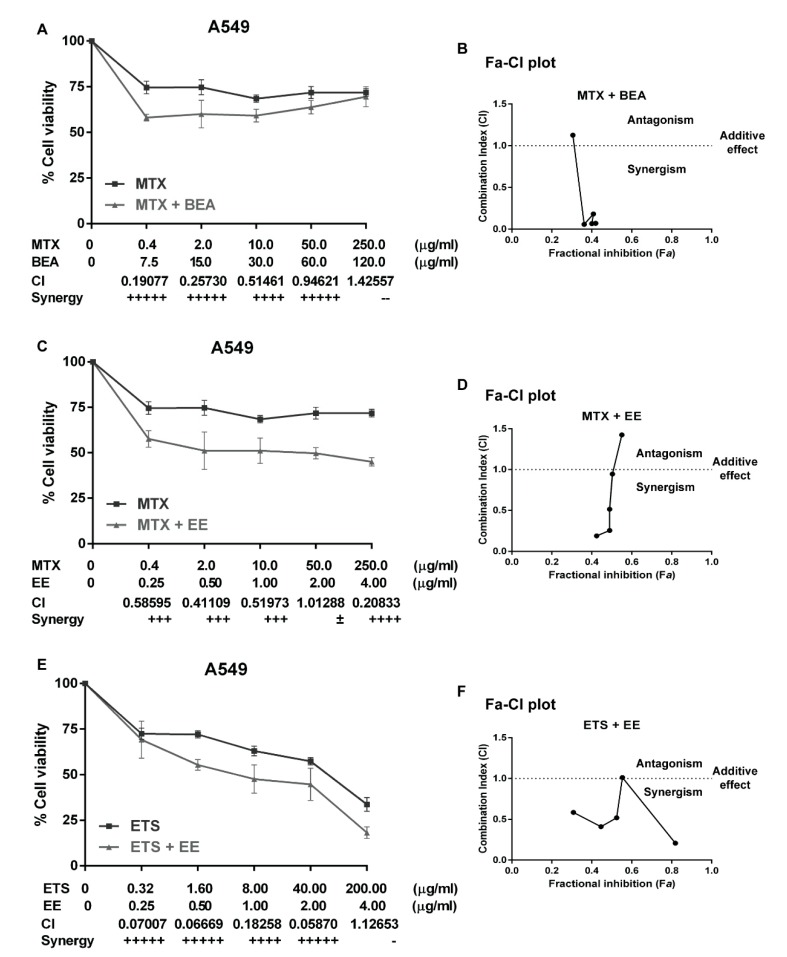
The synergistic combination of chemotherapy drugs with the effective extracts against A549 cells. (**A**,**C**,**E**) The cytotoxicity plots show the percentage of cell viability of A549 cells on the *Y*-axis. The cells were treated with MTX or ETS alone, and combination of MTX + BEA, MTX + EE, and ETS + EE. The concentration of MTX, ETS, BEA, and EE were indicated at the *X*-axis which also showed CI values and their synergy interpretation of each combined concentration. The explanation of the synergy level and symbol are showed in the Table 6. The data are reported as mean ± SD of three independent experiments and was carried out in triplicate. (**B**,**D**,**F**) The combination index (CI) plots correspond to the cell viabilities of each combination and were interpreted by CompuSyn software analysis.

**Figure 5 molecules-25-00231-f005:**
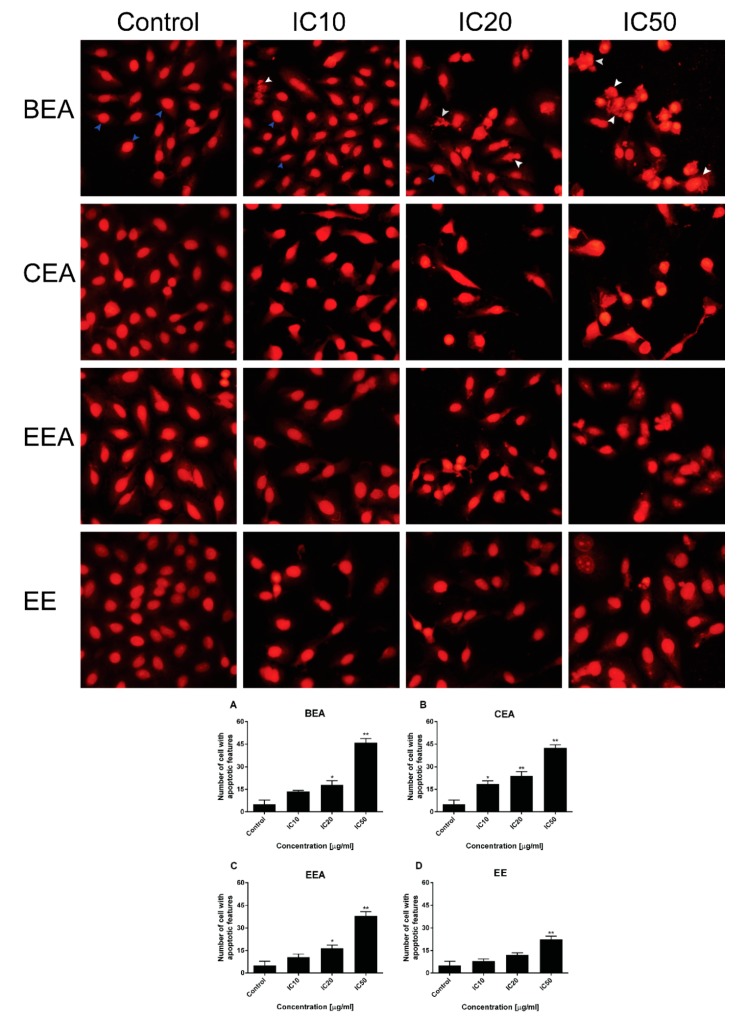
The effective extracts induced an A549 cell apoptotic feature. Morphology of the A549 cells was determined and quantified after treatment with (**A**) BEA, (**B**) CEA, (**C**) EEA, and (**D**) EE for 24 h. Apoptotic bodies and condensed nuclei appeared after staining with propidium iodide. White arrows represent apoptotic cells, whereas blue arrows represent non-apoptotic cells. The number of cells with an apoptotic feature was shown in the bar graphs for 200 total cells counting. Significant results were compared with the control (without treatment) and are shown by * *p* < 0.05, ** *p* < 0.01.

**Figure 6 molecules-25-00231-f006:**
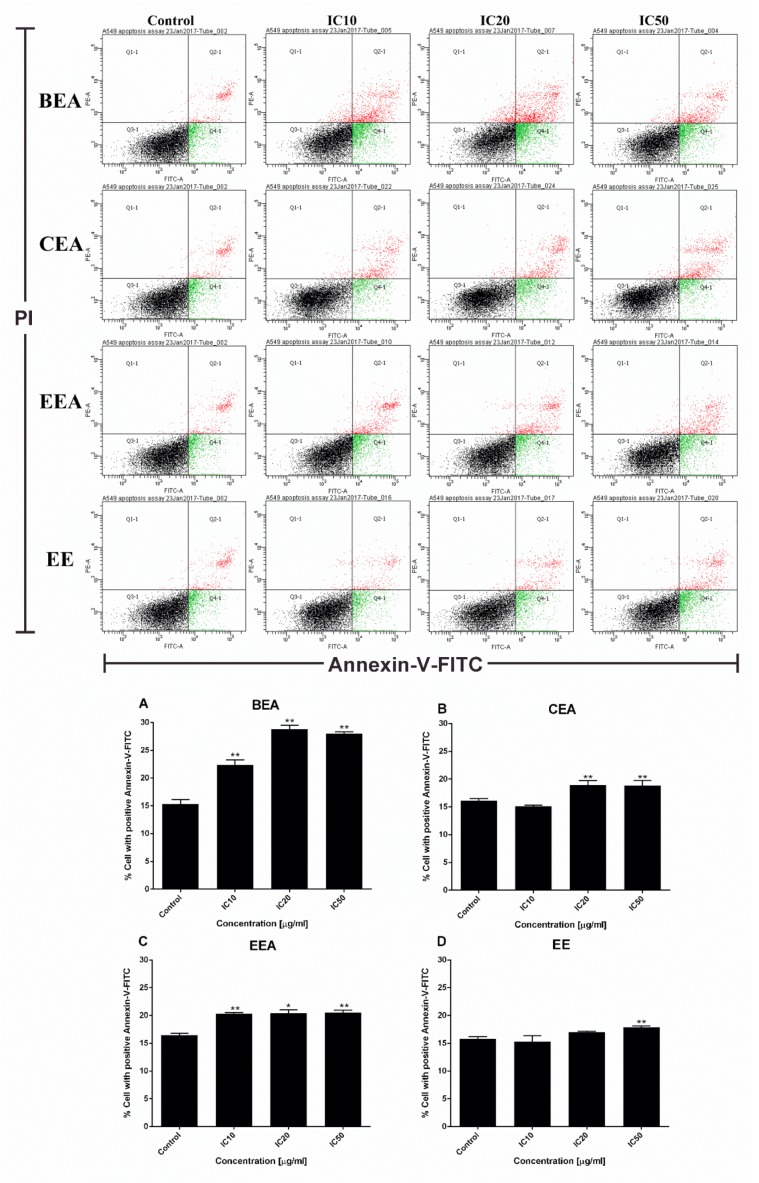
Effective extracts induced A549 cell apoptosis detected by flow cytometry. A549 cells were stained with annexin V-FITC and PI after treatment with the effective extracts for 24 h. Dot plots show annexin V-FITC+/PI− as early apoptotic cells and annexin V-FITC+/PI+ as late apoptotic cells. Early and late apoptotic cells accumulated as a percentage of cell apoptosis in the bar graphs of each extract; (**A**) BEA, (**B**) CEA, (**C**) EEA, and (**D**) EE. The data were reported as a mean ± SD of three independent experiments carried out in triplicate, * *p* < 0.05 and ** *p* < 0.01 compared to the control condition.

**Figure 7 molecules-25-00231-f007:**
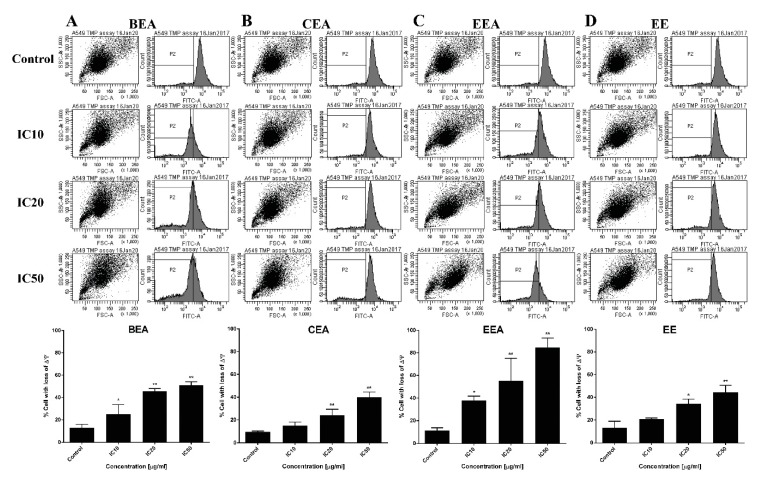
The effective extracts induced ∆Ψm reduction in A549 cells. Histograms (upper) show the reduced fluorescence intensity of DiOC_6_ after treatment with the effective extracts. The percent of the cells with ∆Ψm loss after cells which were treated by (**A**) BEA, (**B**) CEA, (**C**) EEA, and (**D**) EE are shown on the bar graphs (lower). The data are the mean ± SD of three independent experiments carried out in duplicate, * *p* < 0.05 and ** *p* < 0.01 with respect to the control condition.

**Figure 8 molecules-25-00231-f008:**
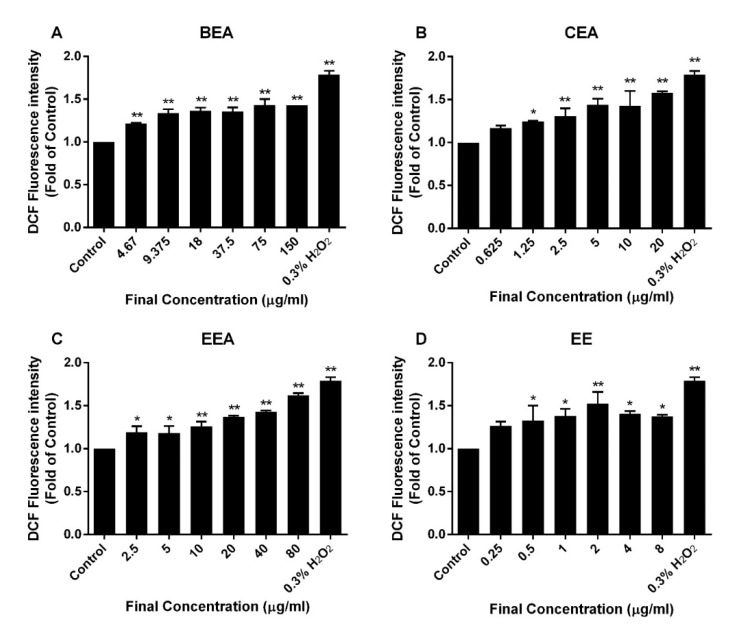
ROS generation in A549 cells after treatment with the effective extracts. The level of intracellular ROS was determined by DCFH-DA fluorescence dye. The increase of fluorescence intensity, which indicates that ROS are generated, is shown on the bar graphs after treatment with (**A**) BEA, (**B**) CEA, (**C**) EEA, and (**D**) EE for 4 h. H_2_O_2_ 0.3% was used as a positive control for ROS production. The data were a mean ± SD of three independent experiments carried out in duplicate, * *p* < 0.05, ** *p* < 0.01 with respect to the control condition.

**Figure 9 molecules-25-00231-f009:**
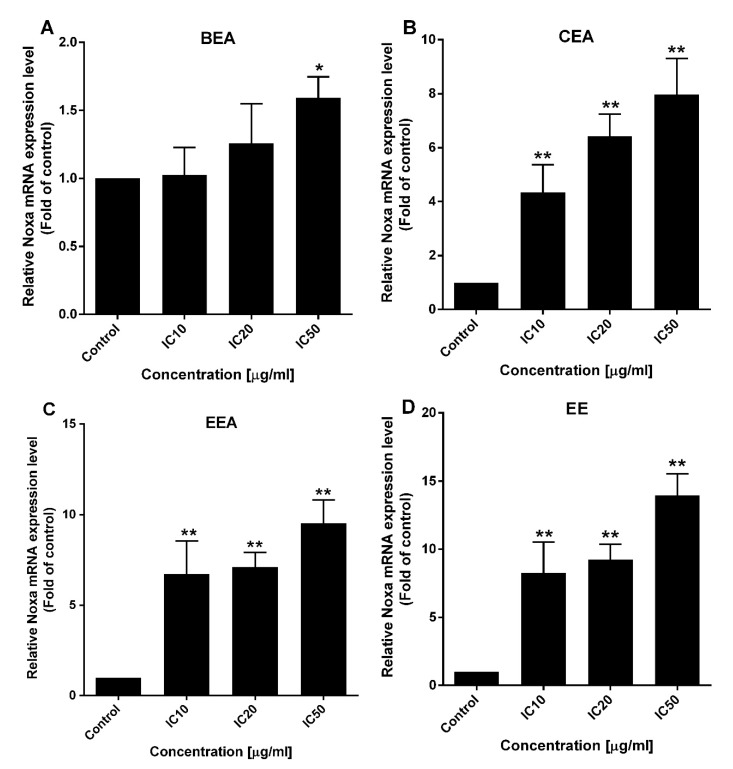
The effect of the effective extracts on *Noxa* mRNA expression level. Quantitative RT-PCR of the mRNA expression of *Noxa* mRNA was measured by a 7500 Fast Real-time PCR system. A549 cells were treated with (**A**) BEA, (**B**) CEA, (**C**) EEA, and (**D**) EE at the indicated concentrations. The relative gene expression level was analyzed by 2^−∆∆Ct^, using *GAPDH* as a housekeeping gene. * *p* < 0.05, ** *p* < 0.01, treatment group vs. control un-treatment groups.

**Figure 10 molecules-25-00231-f010:**
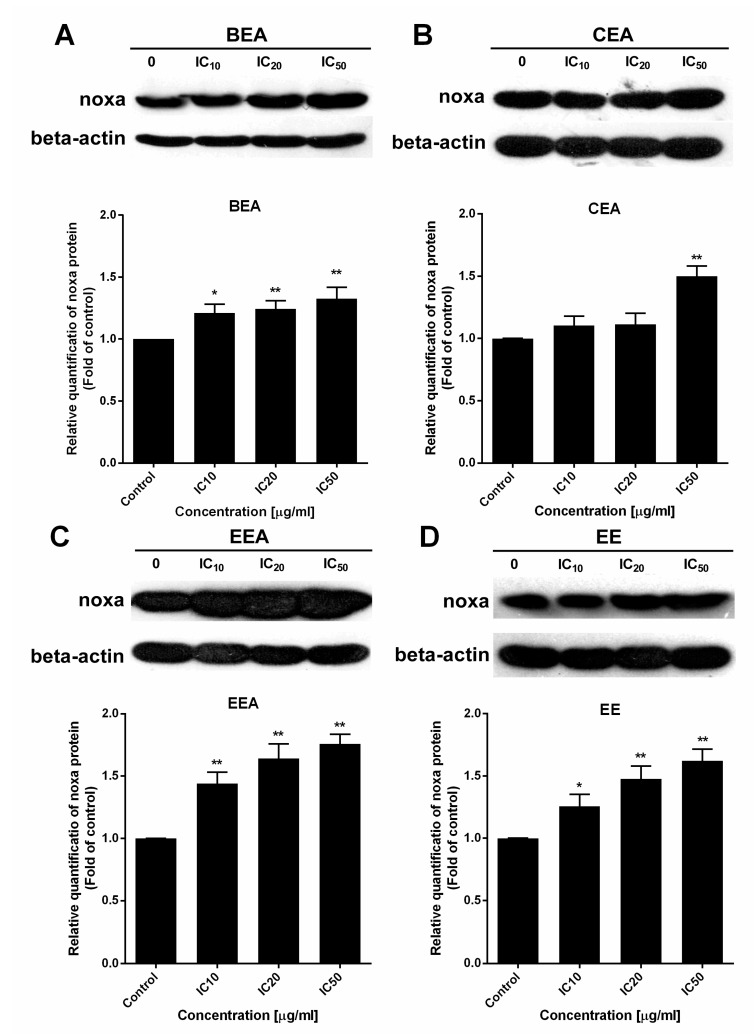
The effect of the effective extracts on Noxa protein expression level. Representations of Noxa protein bands after A549 cells were treated with (**A**) BEA, (**B**) CEA, (**C**) EEA, and (**D**) EE at indicated concentrations as shown in different lanes. The density of each band was measured and calculated for the relative protein level using beta-actin as an internal control (bar graphs). * *p* < 0.05, ** *p* < 0.01, treatment group vs. control un-treatment groups.

**Table 1 molecules-25-00231-t001:** The IC values of the effective extracts at 24-, 48-, and 72-h incubation time against A549 cells and PBMCs, and SI of each extract.

	BEA (µg/mL)	CEA (µg/mL)	EEA (µg/mL)	EE (µg/mL)
24 h: A549: IC_10_	8.84 ± 3.58	0.30 ± 0.13	2.75 ± 0.13	0.19 ± 0.03
IC_20_	14.29 ± 4.63	0.69 ± 0.28	5.79 ± 0.25	0.37 ± 0.06
IC_50_	28.84 ± 5.14	4.12 ± 0.10	18.45 ± 1.40	1.02 ± 0.16
PBMCs: IC_50_	>200	118.41 ± 8.07	>200	>200
SI *	>6.93	28.74	>10.84	>196.07
48 h: A549: IC_50_	22.02 ± 2.34	0.86 ± 0.18	9.08 ± 1.06	0.89 ± 0.26
PBMCs: IC_50_	>200	51.25 ± 9.41	>200	125.92 ± 17.15
SI	>9.08	59.59	>22.02	141.48
72 h: A549: IC_50_	17.39 ± 1.88	0.66 ± 0.01	12.38 ± 2.57	0.96 ± 0.14
PBMCs: IC_50_	>200	41.09 ± 5.82	>200	133.70 ± 12.43
SI	>11.50	62.26	>16.16	139.27

* Selectivity index (SI) is the ratio between IC_50_ of PBMCs and that of A549 cells.

**Table 2 molecules-25-00231-t002:** Retention time, name, and percentage of the total of the enabling identification compounds from BEA.

No.	Retention Time	Name of Compounds	% of Total
1	20.934	Pyrogallol	3.09
2	25.144	Elemicin	0.44
3	30.091	Myristic acid	0.52
4	31.831	Oxacyclotetradecane-2,11-dione, 13-methyl-	0.34
5	34.432	Palmitic acid	9.09
6	36.075	Heptadecanoic acid	0.34
7	37.589	Linoleic acid	0.60
8	37.715	Petroselinic acid	2.28
9	38.042	Stearic acid	1.09
10	43.913	1,2-Benzenedicarboxylic acid	0.29
11	48.047	Squalene	0.24
12	49.294	1-Nonadecene	0.44
13	52.003	1-Tricosene	2.36
14	52.195	alpha-Tocopherol	0.86
15	52.921	Cirsilineol	0.61
16	53.395	Campesterol	3.74
17	53.937	Stigmast-22-en-3-ol, (3α,5α,22E,24ξ)	0.24
18	54.400	Hebesterol	1.11
19	54.768	Stigmast-5-en-3-ol	22.52
20	54.889	Fucosterol	1.23
21	56.010	Stigmasta-3,5-dien-7-one	0.98
22	56.687	Stigmast-4-en-3-one	0.81
23	56.855	Citrostadienol	0.81
24	58.042	Epifriedelanol	1.37
25	58.610	Friedelan-3-one	28.99
26	59.809	2,7-dihydroxy-5-methoxy-3-methylanthraquinone	2.51

**Table 3 molecules-25-00231-t003:** Retention time, name, and percentage of the total of the enabling identification compounds from CEA.

No.	Retention Time	Name of Compounds	% of Total
1	15.676	Benzoic acid	3.24
2	34.197	Palmitic acid	1.75
3	35.792	Kaur-16-ene	1.31
4	38.264	2,3,3-Trimethyl-2,3-dihydro-4h-furo [3,2-C][1]benzopyran-4-One	1.26
5	38.370	Androst-5-en-3beta-ol	1.80
6	42.429	Benz [a] anthracenone	0.14
7	42.847	Kaur-16-en-18-oic acid	50.24
8	43.140	(4-beta)-Kaur-16-en-18-oic acid	2.28
9	49.731	Aflatoxin G1	1.40
10	53.385	Campesterol	0.98
11	53.740	Stigmasterol	2.06
12	54.611	Stigmast-5-en-3-ol	3.60
13	55.993	Stigmasta-3,5-dien-7-one	2.61

**Table 4 molecules-25-00231-t004:** Retention time, name, and percentage of the total of the enabling identification compounds from EEA.

No.	Retention Time	Name of Compounds	% of Total
1	18.623	Resorcinol	2.18
2	21.056	Pyrogallol	11.73
3	21.488	Vanillin	0.27
4	24.194	2,4-Di-tert-butylphenol	0.30
5	34.500	Palmitic acid	12.89
6	37.941	Oleic acid	22.74
7	38.192	Stearic acid	1.08
8	43.932	1,2-Benzenedicarboxylic acid	0.58
9	46.779	beta-Glyceryl monostearate	0.50
10	48.057	Squalene	0.30
11	52.211	alpha-Tocopherol	1.70
12	53.418	Campesterol	2.92
13	53.549	14-Methylergosta-8,24(28)-dien-3-ol	0.26
14	53.782	Stigmasterol	4.24
15	54.360	4,14-Dimethylergosta-8,24(28)-dien-3-ol	0.69
16	54.469	5,8-Dimethoxy-2-methyl-4H-naphtho[2,3-b] pyran-4,6,9-trione	0.54
17	54.672	Stigmast-5-en-3-ol	4.14
18	54.784	beta-Amyrone	1.90
19	55.532	alpha-Amyrone	0.42
20	56.488	24-Methylenecycloartan-3-one	1.70
21	56.858	24-Methylenecycloartanol	1.72
22	59.985	Octadecyl 3-(3,5-di-tert-butyl-4-hydroxyphenyl) propionate	0.60

**Table 5 molecules-25-00231-t005:** Retention time, name, and percentage of the total of the enabling identification compounds from EE.

No.	Retention Time	Name of Compounds	% of Total
1	16.084	Pyrocatechol	1.40
2	16.955	5-Hydroxymethyl-2-farancarboxaldehyde	2.18
3	18.711	Resorcinol	6.26
4	19.134	2-Methoxy-4-vinylphenol	0.82
5	20.125	2,6-Dimethoxyphenol	0.34
6	21.253	Pyrogallol	37.82
7	24.891	4-Hydroxy-3-methoxyphenethanol	1.33
8	25.877	Benzoic acid	0.58
9	26.632	3,4,5-Trimethoxyhenol	0.99
10	34.207	Ethyl gallate	21.75
11	34.721	Ethyl hexadecanoate	1.34
12	37.405	Linoleic acid	0.90
13	37.545	Oleic acid	3.15
14	37.791	Ethyl linoleate	1.18
15	37.918	Ethyl oleate	2.44
16	45.741	2,6-Diphenylimidazo[1,2-b]-[1,2,4]-triazine	1.94
17	46.769	beta-Glyceryl monostearate	2.03

**Table 6 molecules-25-00231-t006:** The classification of the combination index (CI), their indication, and symbols in drug combination studies analyzed with the CI method which is based on those described by Chou and Talalay and the CompuSyn software of Chou and Martin.

CI Range	Description	Symbol *
<0.10	Very strong synergism	+++++
0.10–0.30	Strong synergism	++++
0.30–0.70	Synergism	+++
0.70–0.85	Moderate synergism	++
0.85–0.90	Slight synergism	+
0.90–1.10	Nearly additive	±
1.10–1.20	Slight antagonism	-
1.20–1.45	Moderate antagonism	--
1.45–3.30	Antagonism	---
3.30–10.00	Strong antagonism	----
>10.00	Very strong antagonism	-----

* The symbol is demonstrated the level of synergism or antagonism and mentioned in the Figure 4.

**Table 7 molecules-25-00231-t007:** Patient characteristics (nine patients).

Variables	N or Mean ± SD
Age	63.00 ± 6.12
Gender	
Female	7
Male	2
Success culture	
Cancer cells	9
Normal cells	7
Histology	
Adenocarcinoma	7
Squamous cell carcinoma	2
Cell differentiation	
Well	4
Moderate	2
Severe	3
Staging (8th edition of TNM)	
IA3	2
IB	1
IIB	4
IIIA	1
IIIB	1
Tumor size, mm	55 (35–70)
Intratumoral lymphatic invasion	8
Intratumoral vascular invasion	2
Tumor necrosis	3
Molecular testing	
*EGFR* * mutation	
Wildtype	3
Exon 19 deletion	1
Exon 21 L858R	1
Exon 19 deletion and Exon 21 L858R	1
Unknown	3

* EGFR = Epidermal growth factor receptor gene.

**Table 8 molecules-25-00231-t008:** IC_50_ of each substance (extracts or drugs) and *p*-value compared the IC_50_ or sample number between lung cancer samples and normal lung samples.

Substances	IC_50_, Mean ± SD or Median (Interquartile Range) *		*p*-Value
	Cancer Samples (*n* = 9)	Normal Samples (*n* = 7)	
**BEA (µg/mL)**	139.10 ± 34.60	143.78 ± 75.85	0.901 *
Sensitive cell (n)	6	2	0.315 **
Resistant cell (n)	3	5	
**CEA (µg/mL)**	127.79 ± 54.25	132.37 ± 9.52	0.833
Sensitive cell (n)	9	6	0.437
Resistant cell (n)	0	1	
**EEA (µg/mL)**	60.79 (35.61–120.52)	47.98 (44.33–60.66)	0.775
Sensitive cell (n)	7	6	1.000
Resistant cell (n)	2	1	
**EE (µg/mL)**	8.26 (4.05–13.15)	16.39 (13.27–18.05)	0.013 ***
Sensitive cell (n)	9	6	0.437
Resistant cell (n)	0	1	
**Doxorubicin (µM)**	3.54 (1.28–7.53)	2.36 (1.13–12.25)	0.886
Sensitive cell (n)	6	7	0.212
Resistant cell (n)	3	0	
**Etoposide (µM)**	13.36 ± 4.63	11.96 ± 5.21	0.632
Sensitive cell (n)	6	6	0.585
Resistant cell (n)	3	1	
**Vinblastine (nM)**	11.79 (6.36–12.68)	9.20 (4.16–17.01)	0.775
Sensitive cell (n)	6	7	0.212
Resistant cell (n)	3	0	

* Normal distribution data were reported the IC_50_ as mean ± SD and *p*-values were calculated by unpaired *t*-test. Whereas, non-normal distribution data were reported the IC_50_ in median (interquartile range) and *p*-values were determined by the Wilcoxon rank-sum test. ** *p*-values of the sensitive cell were calculated by Fisher exact test for comparing the proportions of the number of cancer and normal in sensitive group. *** *p* < 0.05; significant at 95% confidence interval.

**Table 9 molecules-25-00231-t009:** Mean difference of IC_50_ compared to cancer samples and normal samples, 95% confidence interval and *p*-value.

	Mean Difference *	95% Confidence Interval	*p*-Value
BEA	+63.93	−21.05, +148.92	0.140
CEA	+0.87	−41.70, +43.43	0.968
EEA	+6.98	−32.39, +46.34	0.728
EE	−4.38	−8.57, −0.18	0.041 **
Doxorubicin	−3.39	−9.50, +2.72	0.277
Etoposide	+0.74	−6.49, +7.97	0.841
Vinblastine	−5.24	−11.69, +1.21	0.112

* Mean difference was analyzed by multi-level random effect model in which IC_50_ values were adjusted by age, gender, cell type, and tumor size. ** *p* < 0.05; significant at 95% confidence interval.

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
