# Peer review of "Potential of Thai Herbal Extracts on Lung Cancer Treatment by Inducing Apoptosis and Synergizing Chemotherapy"

_molecules, 2020, doi:10.3390/molecules25010231_

Round 1
Reviewer 1 Report
In the part 2.1, the authors should provide a time-dependent responses of cancer and normal cells to the herbal extracts. In the part 2.2, the concentrations of herbal extracts and chemo drug for combination treatment were not properly paired. Herbal extracts and chemo drug should have 4 or more concentrations (including 0) per each to give 16 or more pairs of combination. In the part 2.3, it is better using HPLC-MS to identify the chemical constituents in herb extracts considering the limited detecting potential of GC/MS. In addition, it seems that the herbal extracts were not methylated before running GC/MC? Please confirm. BE and CE should also run a GC/MS. The morphology of apoptotic bodies and condensed nuclei in Fig 5 is not clearly shown. The authors should provide magnified figures. Fig 9, Y-axis should be “Relative Nova mRNA expression level”. The title of Fig 9 should be “The effect of the effective extracts on noxa mRNA expression level”. L261-L264, “The cell samples that were reduced the percentage of cell viability to less than 50 are considered to be the sensitive samples for each substance treatment. On the other hand, cell samples which possessed cell viability over 50% were identified as the resistant samples.” Definition of sensitive or resistant cells to the treatment like this is not reasonable. To separate the sensitive and resistant cells, the key should be how much different in the cytotoxicity of extracts between sample cells instead of 50% viability as a cutoff value. In Table 8, how to calculate the p value when the samples are less than 3 in cancer or normal group?Author Response
Please see the attachment.
Thank you.

Reviewer 2 Report
The paper of R. Banjerdpongchai and co-authors is an interesting study on three plant extracts prodrug potential. The extracts were characterized with modern analytical methods; the in vitro/ ex vivo testing featured several aspect of their antitumor potential, and the data were analyzed statistically, generating a large scale of data. Still, the manuscript needs some improvements, such:
Introdution chapter: - reference 1 should be replaced or an additional, newer reference should be cited for updated statistics.
-row 54: please reformulate: "Moreover, chemotherapy drug resistance is common in advanced lung cancer stage", since several stages are considered as advanced in lung cancer, and the chemoresistance towards a certain drug or drug combination could occur in early stages as well.
- row 58: please replace "chemo drugs" with a more academic formulation
Results: - I suggest to place chapter 2.3 ahead, because it contains the characterization of the extracts.
Row 102: please specify the duration of the cells treatment ( ex. 24 hours)
- Row 110- the abbreviation PBMC appears first time in row 110 in the manuscript, therefore the meaning should be inserted here and not in the next paragraph.
In the Methods section they are no details about the approvals from the ethical committee of the host institution, and the informed consent of the donor since its blood sample was used for research purposes.
- Why only E. succirubrum was prepared as ethanolic extract, besides the ethyl acetate extract?
- Figure 3, synergy : why the authors used as reference the drugs etoposide and vinblastine, which are not FDA-approved drugs for non-small cell cancer (the model, A459 is a non-small cancer originated cell line), and methotrexate, approved, but uncommonly used in this pathology. The major active metabolite of methotrexate, 7-OH-MTX is active against cancer, therefore in vitro it should be tested the metabolite, not the drug itself. I recommend to add a synergy testing with the drugs cisplatin/carboplatin, gemcitabine or paclitaxel, and at least doxorubicin, used in Chapter 2.8.
-Table 2: which studied drug combinations correspond to the described synergy levels? A resume of the natural extract-drug interactions could be useful, together with the matematical parameters of Table 2.
- please specify if the apoptosis testing, the transmembrane potential reduction and the ROS level, gene expression were measured after the same treatment extent (an identical with MTT testing or other time interval?).
- chapter 2.8., cell resistance: "The sensitive sample and resistant sample were separated by the percentages of cell variability. The cell samples that were reduced the percentage of cell viability to less than 50 are considered to be the sensitive samples for each substance treatment. On the other hand, cell samples which possessed cell viability over 50% were identified 263 as the resistant samples."- a cells were not characterized with molecular methods to prove their chemoresistance. The enrollment of some samples as "resistant" was made following an algorithm published elsewhere or it is an arbitrary classification?
- It could be useful to identify possible correlations (if any) between IC50, apoptosis induction, ROS level, NOXA expression or other parameters.
Round 2
Reviewer 1 Report
As GC/MS can only detect a portion of the phytochemicals, the authors should be careful when discuss the connections of detected compounds with anticancer activity of the herb extracts. The morphology of apoptotic bodies and condensed nuclei in magnified Fig 5 is still not clear. Please use high resolution pictures if any. It is better to calculate Selectivity Index (SI) by comparing 48-h or 72-h cytotoxicity of herb extracts in cancer cells and PBMCs, or list all SI of 3 time points. “at the time dependent manner” should be “in a time dependent manner”.
Author Response
1. As GC/MS can only detect a portion of the phytochemicals, the authors should be careful when discuss the connections of detected compounds with anticancer activity of the herb extracts.
The discussion parts of GC-MS results was edited and removed some sentence that might be too excessive interpretation and opinion to point out the results of anticancer activity of the extracts. Thank you so much for your concern and advice, adding the 7th and 13th paragraphs and deletion some sentences of the result interpretations in paragraphs 4, 5 and 6, as in tract changes, etc.
2.The morphology of apoptotic bodies and condensed nuclei in magnified Fig 5 is still not clear. Please use high resolution pictures if any.
Thank you for your suggestion. Figure 5 was edited by cropping the fluorescence images to see the cells closer and magnifying some brightness, contrast, and exposure. The .tiff file (dpi 600) was attached, in case the editor has to use high resolution figure.
3.It is better to calculate Selectivity Index (SI) by comparing 48-h or 72-h cytotoxicity of herb extracts in cancer cells and PBMCs, or list all SI of 3 time points.
The table 1 was edited by adding IC50 of A549 and PBMCs, and SI at 48 and 72 hrs. Thank you so much.
4.“at the time dependent manner” should be “in a time dependent manner”.
Thank you for your advice, the sentence was edited as the track changes.